# Citicoline: A Superior Form of Choline?

**DOI:** 10.3390/nu11071569

**Published:** 2019-07-12

**Authors:** Kamil Synoradzki, Paweł Grieb

**Affiliations:** Department of Experimental Pharmacology, Mossakowski Medical Research Centre, Polish Academy of Sciences, 5 Pawinskiego Street, 02-106 Warsaw, Poland

**Keywords:** citicoline, choline, health claims, toxicity, trimethylamine oxide, procognitive effects

## Abstract

Medicines containing citicoline (cytidine-diphosphocholine) as an active principle have been marketed since the 1970s as nootropic and psychostimulant drugs available on prescription. Recently, the inner salt variant of this substance was pronounced a food ingredient in the major world markets. However, in the EU no nutrition or health claim has been authorized for use in commercial communications concerning its properties. Citicoline is considered a dietetic source of choline and cytidine. Cytidine does not have any health claim authorized either, but there are claims authorized for choline, concerning its contribution to normal lipid metabolism, maintenance of normal liver function, and normal homocysteine metabolism. The applicability of these claims to citicoline is discussed, leading to the conclusion that the issue is not a trivial one. Intriguing data, showing that on a molar mass basis citicoline is significantly less toxic than choline, are also analyzed. It is hypothesized that, compared to choline moiety in other dietary sources such as phosphatidylcholine, choline in citicoline is less prone to conversion to trimethylamine (TMA) and its putative atherogenic N-oxide (TMAO). Epidemiological studies have suggested that choline supplementation may improve cognitive performance, and for this application citicoline may be safer and more efficacious.

## 1. Introduction

Citicoline is the international nonproprietary name (INN) for cytidine-diphosphocholine (CDP-Cho). The substance is commercially available in two forms, sodium salt and inner salt. Citicoline sodium salt, classified as a nootropic and psychostimulant [1], is an active principle of a variety of prescription drugs, either injectables or oral formulations. In 2009 in the USA, citicoline (inner salt) was self-affirmed by the Japanese company Kyowa-Hakko as GRAS (generally regarded as safe) [2], and in 2014 it was announced as a novel food ingredient by the appropriate Implementing Decision of the Commission of the European Union [3].

The aforementioned EU Implementing Decision states that citicoline may be placed on the EU market, where it is intended to be used in food supplements aimed at a target population of middle-aged to elderly adults at a maximum level of 500 mg/day, and in dietary foods for special medical purposes with a maximum dose of 250 mg per serving and with a maximum daily consumption level of 1000 mg from these types of foods.

## 2. Citicoline in Food Supplements: The Issue of Health Claims

Classifying citicoline as a food ingredient suitable for food supplements should make it widely available, but in the highly regulated market of the European Union its marketing is problematic. According to the EU Regulation EC No 1924/2006 [4], all nutrition and health claims made in commercial communications concerning food supplements must be formally authorized following scientific assessment performed by the European Food Safety Agency (EFSA). Citicoline does not have any nutrition or health claim authorized up to date. Moreover, application for authorization of a health claim (related to citicoline and maintenance of normal vision) was turned down by the EFSA because it was concluded that a cause and effect relationship has not been established between the consumption of citicoline and the maintenance of normal vision [5]. Does this mean that, although it is legal to introduce citicoline to the EU market in a food supplement, information provided about this supplement should not contain any information about its specific nutritional and/or functional value?

Looking through the positive EFSA Scientific Opinion on citicoline issued prior to the aforementioned implementing decision [6], we find the reference to the observation that, both in humans and in rats, upon ingestion citicoline undergoes quick hydrolysis, breaking down to choline and cytidine [7], which then undergo further metabolism and incorporation into normal pathways of metabolism [8]. Cytidine, a pyrimidine nucleoside which in humans interconverts with uridine [9], undergoes intracellular phosphorylations to cytidine triphosphate (CTP), which participates in phospholipids synthesis via the Kennedy pathway, and may also be incorporated into nucleic acids. Choline is either phosphorylated to phosphocholine and participates in phosphatidylcholine synthesis, or oxidized to betaine, which serves as a methyl donor in the betaine-homocysteine methyltransferase reaction. Also, in cholinergic neurons, choline is acetylated to form the neurotransmitter acetylcholine.

We may, therefore, consider citicoline as a source of choline and cytidine. Whereas there is no nutrition or health claim authorized for cytidine either, there are three such claims authorized for choline. These are so-called functional claims relating to the beneficial effects of a nutrient on certain normal bodily functions. The first two state that choline contributes to normal lipid metabolism and to the maintenance of normal liver function. These claims were accepted because they were substantiated by observations that choline deficiency is associated with signs of liver damage (elevated serum alanine aminotransferase activity) and the development of fatty liver (hepatosteatosis) in humans fed choline-free total parenteral nutrition solutions, whose effects can be reversed by the administration of dietary choline [10,11]. The third claim, stating that choline contributes to normal homocysteine metabolism, was substantiated by the observations that choline-depleted diets tend to increase plasma concentrations of homocysteine [12], whereas human observational [13,14] as well as intervention [15] studies supported the inverse association between dietary choline and blood concentrations of homocysteine. Of note is that in the aforementioned intervention study, choline was supplied orally in the form of phosphatidylcholine (lecithin).

At the same time, health claims stating that choline contributes to the maintenance of normal neurological function and normal cognitive function were rejected by the EFSA because cause and effect relationships have not been established between the consumption of choline and the claimed effects [16]. One of the reasons was that some references that presented support for the claimed effects described studies that did not evaluate choline, but, for example, citicoline. A possible explanation of this paradox is that at the date of issuing scientific opinion on the health claims concerning choline (i.e., year 2011), citicoline was not yet appreciated by EFSA experts as the dietary source of choline. Indeed, natural foods do not contain any significant amount of this substance.

There is no direct proof that citicoline intake can reverse either elevated serum alanine aminotransferase activity or the development of fatty liver in people who are choline-deficient. There is also no direct proof that citicoline intake may lower homocysteine in blood. On the contrary, single oral administration of a high dose of citicoline (1 g/kg b.w.) to rats resulted in a transient increase of plasma homocysteine, but when a lower dose was supplemented in the diet for two months, plasma homocysteine remained unchanged [17]. At the same time there is no reasonable doubt that oral intake of citicoline is a safe and efficient method of delivery of choline to the human body.

It might perhaps be concluded that the issue of the applicability to citicoline of health claims pertaining to choline (and apparently also to some of its derivatives, such as phosphatidylcholine) is merely a legal problem that shall be settled accordingly by the appropriate authorities. On the other hand, a health claim authorized almost a decade ago may not be supported in its entirety by the contemporary scientific data. Current guidelines for the management of fatty liver do not mention supplementation with choline or its derivatives [18]. Likewise, folic acid, vitamin B_6_, vitamin B_12_, and betaine, but not cholines, are listed among nutrients that may counteract hyperhomocysteinemia [19].

## 3. Citicoline as a Source of Choline: The Issue of Acute Toxicity

It is well established that following ingestion citicoline is fully absorbed and catabolized to cytidine and choline, which enter their respective metabolic pools in the body [20,21,22]. However, the particulars of its absorption, hydrolysis, and dephosphorylation(s) are a bit unclear. Citicoline contains equimolar amounts of choline and cytidine. Following citicoline ingestion in rats, the increase in both plasma cytidine and choline occurred quickly, but the molar increase in plasma choline was markedly smaller [23]. In a human study [24], oral citicoline resulted in increases in plasma choline and uridine that were similar in timing and magnitude, but in the other human study, the increase in plasma choline following citicoline ingestion was biphasic and delayed [25]. It has been suggested that citicoline is absorbed intact and its hydrolysis occurs in the liver and is coupled with a selective withdrawal of choline from blood [26]. Following oral citicoline intake in humans, the quantitative transformation of cytidine to uridine occurring in the intestine or liver was also postulated [24].

Absorption of intact citicoline molecules from the intestine to blood could also be helpful for explaining differences of acute toxicity of citicoline versus choline upon different routes of administration (Figure 1).

The classical measure of acute toxicity is LD_50_, the median lethal dose of the tested compound expressed in milligrams per kilogram body weight. The lower the LD_50_ value, the more toxic the substance. For any route of administration (oral, intraperitoneal, intravenous), the LD_50_ of citicoline is higher than the corresponding LD_50_ of choline, indicating that citicoline is much less toxic than choline. This difference is certainly not unexpected when we consider that the molecular weight of choline moiety (MW = 104) contributes less than 30% to the molecular weight of citicoline (MW = 489), whereas the acute toxicity of cytidine is probably lower than that of choline. However, when we express the aforementioned LD_50_ values on a molar basis, citicoline is still substantially less toxic than choline. The difference in molar toxicity between citicoline and choline is more than 20-fold when the substances are applied intravenously. Apparently intact citicoline molecules do not evoke acute cholinergic toxicity, probably because they are not substrates for acetylcholine synthesis.

When the compounds are given per os, the difference in toxicity is several times lower, but it still is quite significant. Two possible explanations can be proposed for the aforementioned differences. One could be that when cytidine appears in blood concomitantly with choline, it somehow attenuates acute choline toxicity. The other, which seems more plausible, could be that upon oral application choline is not liberated from citicoline in the intestinal lumen, preventing its conversion to TMA. Compared with phosphatidylcholine and other choline derivatives encountered in food (e.g., carnitine, glycerophosphocholine), citicoline may be less prone to enzymatic hydrolysis inside the intestinal lumen because it is the only compound containing pyrophosphate group (it should, however, be noted that according to one study [32], the distribution of radioactivity in tissues, urine, and expired air following oral and intravenous administration of methyl-^14^C-labeled citicoline in rats showed metabolic differences which suggested that the compound is, at least partially, metabolized to TMA prior to its gastrointestinal absorption).

## 4. Does Resistance to Hydrolysis in the Intestine Make Citicoline a Safer Choline Supplement?

The issue of hypothetical citicoline resistance to intraintestinal hydrolysis is of importance when we consider that the intestinal microbiome metabolizes a significant fraction of choline and its derivatives to trimethylamine (TMA), a gaseous metabolite readily taken up and oxidized in the liver to its N-oxide, TMAO.

TMAO has been implicated in the etiology of various diseases, such as kidney failure, diabetes, and cancer [33]. There is a large and growing amount of literature on the atherogenicity of TMAO resulting in increased incidence of myocardial infarction, stroke, or death [34]. A meta-analysis published recently led to the conclusion that higher plasma TMAO correlates with a 23% increase in risk for cardiovascular events and a 55% increase in all-cause mortality [35]. Two recent reports showed that higher TMAO levels were associated with increased risk of first ischemic stroke and worse neurological deficit [36], and that patients suffering from atrial fibrillation who developed cardiogenic stroke displayed approx. 4 times higher TMAO levels in plasma than patients with atrial fibrillation who did not develop stroke [37]. Another recent report suggested a link between TMAO and Alzheimer’s disease [38]. It has even been suggested that supplementation with choline esters prone to be metabolized to TMA and TMAO, such as phosphatidylcholine, may be dangerous to human health [39].

On the other hand, several observations cast doubt on the pivotal role of TMAO in atherosclerosis. First of all, nutritional intakes of TMAO and its precursors do not always correlate with cardiovascular disease risk. For example, high fish intake increases TMA/TMAO while being cardioprotective. Some hypotheses have been proposed recently to resolve this paradox, employing inter alia a phenomenon of reverse causality, a possible role of insulin resistance and diabetes mellitus in activating N-oxidation of TMA, etc. [40].

Nonetheless, many authors still take it as having been proven that TMAO is a causative factor in the development of atherosclerosis and cardiovascular diseases. For example, in a recent review on TMAO and stroke [41], several reports are quoted that show the importance of TMAO as a risk factor and prognostic marker for this disease, and indicate the pathomechanisms involved. These include increased TMAO generation promoting atherosclerosis, platelet activation, and inflammation. The author concludes that TMAO may be a central molecule in the re­lationship of diet, genetics, the gut microbiota, and cardiovascular disease.

It may be concluded that until the place of TMAO in the chain of events leading to cardiovascular diseases and mortality is ultimately clarified, citicoline could be a more reasonable choice than other choline compounds, when choline supplementation is indicated.

## 5. Citicoline: A “Procognitive” Form of Choline

In two population studies, significant associations were found between choline intake or free choline level in blood and the cognitive performance of adult and elderly people. In a community-based population of non-demented individuals (1391 subjects, mean age 60.9 years), higher concurrent choline intake was related to better cognitive performance [42] (Figure 2). In another cross-sectional study (2195 subjects aged 70–74 years), low plasma free choline concentrations were associated with poor cognitive performance [43]. A possible explanation for the effect of choline intake on cognition in adults has been sought in its function as a precursor of phosphatidylcholine (PC), a major constituent of all biological membranes, and acetylcholine, a neurotransmitter involved in cognition [44].

Therefore, it might be expected that supplementation with choline will improve cognitive performance. However, trials in which the effects of oral supplementation of humans with choline or phosphatidylcholine on cognition were investigated yielded mixed, mostly negative results (see [45] and references cited therein). On the other hand, in a recent small placebo-controlled study, adolescent males treated with citicoline showed improved attention and psychomotor speed and reduced impulsivity [46]. In other recent controlled studies, citicoline seemed to be efficacious in adult patients suffering from cognitive impairments, especially of vascular origin [47]. These newer studies corroborated results obtained previously when citicoline as a prescription drug had been tested in several placebo-controlled trials for cognitive impairment due to chronic cerebral disorders in the elderly. The review of those early trials led to the conclusion that there was some evidence of a positive effect of citicoline on memory and behavior in at least the short to medium term [48]. Moreover, it was recently shown that in patients suffering from dementia concomitant oral intake of citicoline improved the efficacy of cholinesterase inhibitors [49,50].

## 6. Conclusions

Altogether, whereas the jury may still be out on the issue whether, or to what extent, citicoline taken orally is metabolized to TMA and TMAO, there are reasons to believe that procognitive effects of citicoline supplementation are superior over those of choline or phosphatidylcholine.

## Figures and Tables

**Figure 1 nutrients-11-01569-f001:**
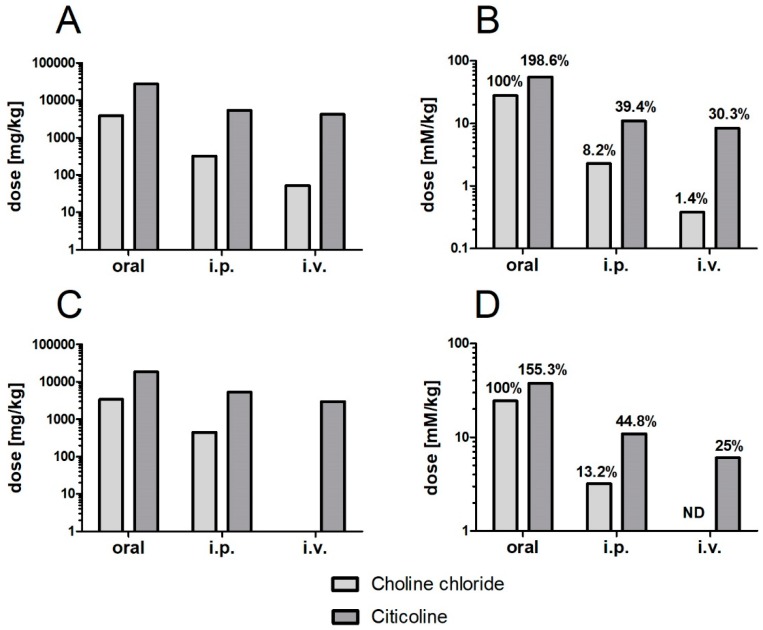
Median lethal dose of choline chloride and citicoline in mice (**A**,**B**) and rats (**C**,**D**) expressed in milligrams (**A**,**C**) or in millimoles (**B**,**D**) per kilogram body weight, depending on the route administration. Data compiled from refs. [27,28,29,30,31]. Abbreviations used: i.v., intravenous; i.p., intraperitoneal; ND, no data available.

**Figure 2 nutrients-11-01569-f002:**
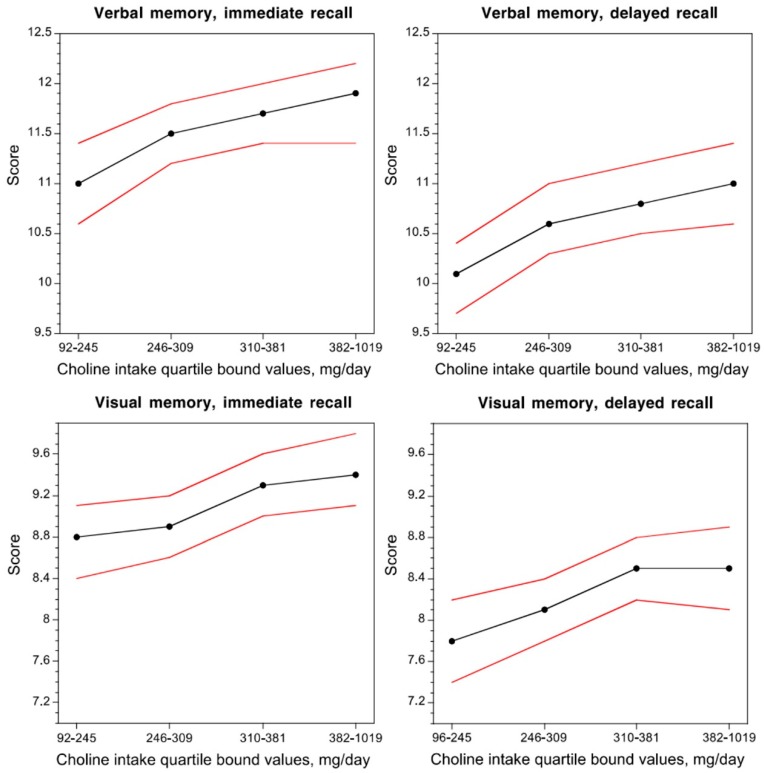
Dose–response relationship between the average daily choline intake and verbal and visual memory performance in non-demented adults. Black lines indicate the mean score and red lines indicate 95% confidence interval. Figure reproduced from [44].

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
