# Peer review of "Citicoline: A Superior Form of Choline?"

_nutrients, 2019, doi:10.3390/nu11071569_

Round 1
Reviewer 1 Report
The paper reports the use of citicoline in food supplements as a source of choline. It is well written. The authors should discuss better the metabolic pathway leading from citicoline to cytidine and choline and then to betaine and homocysteine.
Another important point to be better addressed is that the association of choline and cytidine is able to lower choline toxicity index about 20 fold.
Reviewer 2 Report
For author :
This paper discuss the role of Citicoline as a dietary source of choline in relation with toxicity and potential health claim. Overall, the paper is well written but the key messages don’t come across that clearly in the paper. The abstract is clear but the paper could be improved in that regard.
1- The sections’ title do not always reflect accurately what is discussed in the section. For instance TMAO and cognitive function all under the same title as toxicity…
2- Having 1-2 summary sentence at the end of each section with key point to remember would help to clarify the message.
3- The discussion on TMAO should be expended a little. Yes TMAO has been associated with an increased risk of CVD etc. Some studies have suggested that choline/phosphatidylcholine contribute to plasma TMAO levels but many studies were also unable to confirm this association. Interestingly, fish consumption, which is an important component of the Mediterranean diet (which is cardioprotective), has also been shown to increase TMAO concentrations in plasma more than egg and beef (Cho, C.E.; Taesuwan, S.; Malysheva, O.V.; Bender, E.; Tulchinsky, N.F.; Yan, J.; Sutter, J.L.; Caudill, M.A. Trimethylamine-n-oxide (tmao) response to animal source foods varies among healthy young men and is influenced by their gut microbiota composition: A randomized controlled trial. Mol Nutr Food Res 2017, 61.). I don’t think that there is enough evidence to support a link between dietary choline consumption and TMAO.
4- A conclusion paragraph with a recap of key point would be a good way to end the paper.
5- From my understanding the health benefit associated to citicoline come from its choline content not citicoline.
6- The discussion about the differential toxicity between choline and citicoline and potential explanation are not clear especially the second reason provided line 123-128.
